# Using Quantum Nodes Connected via the Quantum Cloud to Perform IoT Quantum Network

Doaa Subhi * 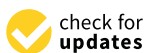 and Laszlo Bacsardi 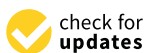

Department of Networked Systems and Services, Faculty of Electrical Engineering and Informatics, Budapest University of Technology and Economics, Műegyetem rkp. 3, H-1111 Budapest, Hungary
* Correspondence: al-nuaimd@edu.bme.hu

**Abstract:** Computer networks consist of millions of nodes that need constant protection because of their continued vulnerability to attacks. Classical security methods for protecting such networks will not be effective enough if quantum computers become widespread. On the other hand, we can exploit the capabilities of quantum computing and communications to build a new quantum communication network. In this paper, we focused on enhancing the performance of the classical client–server Internet application. For this sake, we introduced a novel Internet of Things (IoT) quantum network that provides high security and Quality of Service (QoS) compared with the traditional IoT network. This can be achieved by adding quantum components to the traditional IoT network. Quantum counterpart nodes, channels, and servers are used. In order to establish a secure communication between the quantum nodes and the quantum server, we defined a new Communication Procedure (CP) for the suggested IoT quantum network. The currently available quantum computer has a small qubit size (from 50 to 433 qubits). The proposed IoT quantum network allows us to overcome this problem by concatenating the computation efforts of multiple quantum nodes (quantum processors).

**Keywords:** quantum internet; internet of things; quantum information; quantum networks



## 1. Introduction

The quantum computing age will start soon; it will provide unique properties over the classical ones. Recently, different companies have started businesses in quantum computing. The quantum computing market is expected to grow to more than USD 10 billion by 2024 [1]. It can solve problems exponentially faster than classical computers. Moreover, it can tackle complex mathematical problems by applying quantum algorithms. The most known developed quantum strategies are Shor's algorithm (it factorizes any integer number to its prime number.), the Grover algorithm (it searches for a specific item in an unsorted database, representing a real problem in classical computing) [2], and quantum phase estimation (it computes the phase of the eigenvalue of a unitary operator) [3–7]. It is interesting to note that Shor's algorithm creates a real threat to the existing classical cryptography systems, such as RSA (Rivest–Shamir–Adleman), which is commonly implemented by the classical Internet.

One of the quantum internet protocols is quantum teleportation [8], which performs qubit transmission between two nodes. Quantum teleportation shares quantum entanglement between nodes. If the distance between the two nodes is very large, the shared entanglement between the two nodes will be lost. As an alternative solution, additional quantum components are added between the two nodes in order to conserve the shared entanglement, as so-called quantum repeaters [9]. Quantum communication provides unique properties and applications over classical ones. Highly secure communication can be achieved by applying the quantum key distribution (QKD) cryptographic protocol [10,11]. QKD uses quantum mechanics to securely share random bits for encryption by using a secret key between communicating parties. The key is a string of qubits used to hide the classical messages and could be used for encryption purposes in the classical Internet.

Blind Quantum Computing (BQC) is one of the promising applications of the quantum internet [12,13]. It provides access to the users to perform their computations securely. In the case of sensitive data, the quantum computer can execute the calculation without knowing the type of the content of the data. The special properties of quantum mechanics, such as superposition and entanglement, enable the quantum internet to connect different quantum devices and share information efficiently. It is important to emphasize that the speed of quantum computing increases exponentially as the number of qubits increases.

As the number of IoT devices increases exponentially, efficient quantum algorithms are required to perform the search and optimization. Recently, a considerable number of quantum strategies have been devised [14] that are more efficient than classical ones. Quantum data centers can be accessed via a quantum cloud [13]. Many companies create public quantum clouds allowing researchers to access different numbers of quantum devices, such as IBM company [15], Google [16], etc. It is important to emphasize that the quantum internet will not replace the classical one but will integrate smoothly into the classical framework. This integration between the two types of networks will create a better quality of service for users.

The massive number of IoT devices collect large quantities of data that are sent to the data centers (servers) for processing, making decisions, and sending control signals in order to activate an action [17]. The collected data are very sensitive information. If the information is leaked because of a security system leakage, it may affect the machines/robots used in the factory (for example, losing control of the machine, unexpected shut down in the system, etc.) in the case of industrial application; this will not only influence the factory economy but also will harm people who work in the factory. For this sake, a new alternative secure IoT network is required [18–20]. In this study, we focused on improving the performance of the classical client–server Internet application. To that end, we present a novel IoT quantum network that provides high security and quality of service (QoS) compared with the traditional IoT network. Additionally, we described a new communication procedure for the suggested IoT quantum network. To the best of our knowledge, no IoT quantum scheme has been introduced in the literature. The manuscript is organized as follows: In Section 2, we provide an overview of quantum communication and its related challenges and how the quantum repeater plays a key role in the problem of decoherence, attenuation, and qubit fragility. In Section 3, we present the quantum repeater protocol, set of rules for nodes communication, and quantum network stack, which will be applied in our new IoT quantum network. In Section 4, we suggested a new IoT quantum network, and we defined the quantum procedure communication between the quantum nodes and the quantum server. Section 5 concludes the manuscript.

## 2. Quantum Information Theory

### 2.1. Overview of Quantum Communication

There are essential differences between the capabilities of quantum and classical communication. In quantum information, the smallest building unit is called a qubit, while for classical information it is called a binary digit or bit. Each classical bit only has a single value represented by 0 or 1. A single qubit simultaneously represents 0 and 1 due to the superposition property. The quantum state is defined as a unit vector in Hilbert space. Quantum superposition is a principle of quantum bit that enables it to be represented as a combination of states. The superposition of the quantum state can be expressed by Dirac notation

$$|\psi\rangle = \alpha|0\rangle + \beta|1\rangle, \tag{1}$$

where $|\alpha|^2 + |\beta|^2 = 1$, $\alpha$ and $\beta$ are complex numbers. $|0\rangle$ and $|1\rangle$ are called ket zero and ket one, respectively.

The expression in (1) expresses the superposition property of two states $|0\rangle$ and $|1\rangle$. Due to the superposition principle, quantum computers are powerful, and their speed increases exponentially as the number of qubits increases.

The quantum register principle is based on the superposition principle; it is contained in multi-qubits generated as a result of a tensor product as in Equation (2), which expresses a multi-qubit system (quantum register):

$$\sum_{i=0}^{2^n-1} \alpha_i|i\rangle$$
$$\sum_{i=0}^{2^n-1} |\alpha_i|^2 = 1 \tag{2}$$

Qubits are grouped and indexed in the quantum register. Each qubit indexed with index ($i$) started at 0 and increased by one each time. Complex parameters describe the register system; a two-qubit system is represented by four complex parameters, while eight complex parameters represent a three-qubit system.

Entanglement, an important phenomenon in quantum mechanics, represents the core of the quantum internet. The quantum internet is based on the distribution of entangled states between nodes. Quantum entanglement enables the transmision of unknown states without the need to know their state, thus keeping it secure. This special quantum phenomenon enables particles to share a special relationship that cannot describe the state of the individual particle without describing the state of the other one. To describe the entanglement between two nodes, A and B, the expression in Equation (3) is used.

$$|\psi\rangle = \frac{|00\rangle + |11\rangle}{\sqrt{2}} \tag{3}$$

Quantum teleportation is one of the quantum internet protocols. It transmitted quantum information with the help of pre-shared entanglement between two nodes. The sender and receiver share an EPR pair (see Equation (3)) and then send the unknown qubit to the receiver.

### 2.2. Quantum Communication Limitation

One of the most critical challenges in quantum internet development is the fragility of qubits and their interaction with their surroundings. Quantum information can only travel short distances due to noise and the entanglements of transferred information with the environment. Another challenge is that quantum information cannot be amplified by using the reduplicated method due to the no-cloning theorem [21].

There is another challenge that appears when using a quantum memory chip. A quantum chip has a size similar to the classical one, but special conditions are required to work, such as a particular absolute condition. This qubit must work in a confined space; the interconnecting and preserving of these qubits become more complex as the number of these qubits increases.

### 2.3. Solutions and Possibilities

Quantum repeaters [9] are able to solve the problem of decoherence, attenuation, and qubit fragility. It plays a role similar to the classical one establishing a multi-hop connection and optimal routing selection. Quantum repeaters are based on Bell inequality and entanglement swapping. Bell pairs are generated between neighboring repeaters, and entanglement swapping connects the distance repeaters by connecting their Bell pairs. Quantum repeaters were suggested first by Briegel [22] in 1998. In 2009, Jiang et al. suggested the first quantum repeater protocol that can extend the distance (103–106) Km [23]. This repeater uses Calderbank–Shor–Steane (CSS) code and classical error correction to increase the entanglement fidelity.

A distributed quantum computers network [24] can solve the problem of the few qubits on the present quantum hardware. Such a distributed network will be reliable because it will benefit from the quantum entanglement phenomenon. These quantum

nodes (computers) are connected via a quantum channel. A quantum communication channel enhances the execution of coherent signals by applying M-ary phase shift keying and a probabilistic Noiseless Linear Amplifier (NLAs) [25–27].

## 3. Quantum Protocols and Rules

Quantum internet network success highly depends on the capability to establish communication by using quantum entanglement for long distances. Different architecture have been created since the early 1990s.

In 2016, Jones et al. suggested an analysis of three protocols for quantum repeaters MeetIntheMiddle (MIM), SenderReciver, and Mid-point [28]. Each repeater type has several controllable memory qubits. These memory qubits work to establish memory/photon entanglement. In the first type, the MIM model suggested two repeaters at the end of the link; each repeater sends a photon to the Bell State Analyzer (BSA), which is fitted in the middle of the link via an optical channel as shown in Figure 1. Each of these photons is already entangled with the memory qubit in the repeater node. When the photons reach the BSA nodes and are entangled, the detector connected via the classical channel will inform the repeater node that the entanglement swapping was successful. To establish another entanglement swapping, the repeater node will project memory qubits into the entangled states.

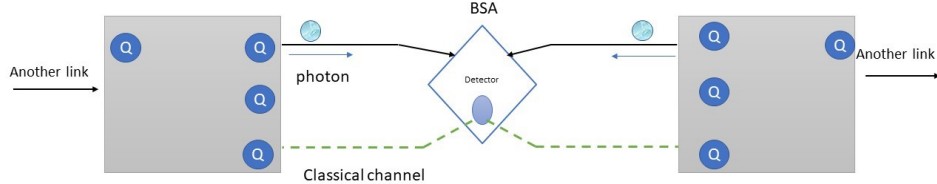

**Figure 1.** MIM quantum repeater model.

In the second protocol, the SenderReciever BSA node will be moved to the receiver side of the repeater, as shown in Figure 2. This protocol works in the same way as the MIM protocol. Still, it will solve the synchronization problem of the first protocol and the decision to establish another entanglement swapping token by the receiving side of the repeater.



**Figure 2.** Sender Receiver quantum repeater model.

The third type, the Mid-point quantum repeater, has more components to ensure no photon loss, as shown in Figure 3. In this type, each repeater node has both BSA and memory qubits and is connected to a source of entangled photons placed in the mid-point on the channel. The source of entangled photons will be generated by entanglement and distributed to both repeaters' nodes. Each repeater node will be able to detect the photons and make decisions.

In 2019, Dahlberg et al. suggested the first quantum network stack and described its functional allocation as shown in Figure 4. It controlled the entanglement distillation [29]. They introduced the first physical and link layer protocol for quantum networks that are able to produce heralding entanglement between quantum processors and provide a reliable service to the connected devices by generating an entangled pair continuously.

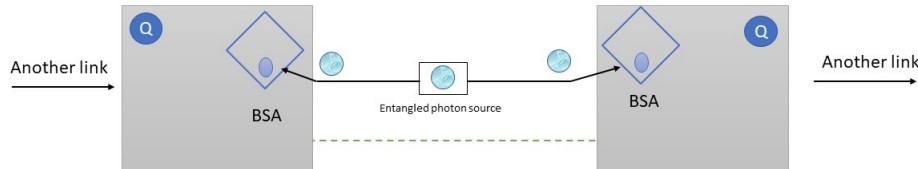

**Figure 3.** Entangled source quantum repeater model.

Takaaki and Rodney Van Meter [30] suggested a quantum rule set (RuleSet) that coordinates quantum operations between multi-nodes over long distances. It consists of a group of conditions and actions. The conditions are clauses that contain a set of conditional statements. At the same time, the actions are operations performed by the node with respect to the conditions, such as entanglement swapping and message generation. Another concept to be defined is the so-called RuleEngine, it refers to a software module installed in each node. It is utilized to manage and execute RuleSet actions. To identify the time of the RuleSet generation and IP address for the node, a RuleSet identifier (RuleSetID) was introduced using the hash function. At the end of each action, a counter called "action indexed" is increased by one. Takaaki and Rodney also adjusted quantum link bootstrapping and simulated it based on the Markov Chain–Monte Carlo simulation model. Moreover, the authors introduced a quantum network interface card (QNIC) that is used to coordinate communication between the quantum nodes.

In 2020, a research group from Keio University, Japan, introduce a quantum internet simulation package (QuISP) [31]. QuISP is a discrete event simulator built on OMNET++; it can simulate a large-scale quantum network and investigate its behavior.

| Layers | Job of each layer | Description |
|---|---|---|
| **Application layer** | | |
| Transport Layer | Qubit transmission | Information transmission |
| Network Layer | Long distance entanglement | End-to-end connections, responsible for long distances nodes that did not connect directly. |
| Link Layer | Producing entanglement | Point-to-point connection (neighboring nodes) |
| Physical Layer | Attempt entanglement generation | Quantum hardware, NV platform, Ion traps, etc, and physical connections such as fiber. |

**Figure 4.** Quantum network stack describing the function of each layer and where quantum entanglement is generated and transmitted.

## 4. IoT Quantum Network Model

The currently available quantum devices have a small qubit size (between 50 and 433 qubits). Connecting a large number of limited qubit size quantum computers/processors via the quantum cloud will allow for providing high computational efforts. Quantum cloud supplies users with resources to process and store huge data. In such a case, users will not necessarily process and store it locally; it could be collected and sent to the server for processing. Next, an action/operation is produced by the server and forwarded back to the output. Using quantum channels to share encryption keys between sender and receiver means the communicating nodes will expose any eavesdropper who tried to attack the network and steal the key.

In this Section, we suggest an IoT quantum network model that employs quantum mechanics properties. These unique properties of quantum computers and quantum channels provide IoT networks with high processing speed and security. The strength and power of quantum computers offer a good choice for use as servers in data centers due to the power provided by quantum registers that can perform computation and process data

with high speed. A quantum search algorithm called the Grover algorithm proved to have a higher speed than the classical one. High security is due to the quantum information properties that humans cannot sense unless measured; when the measurement is applied, the two communication parties perceive an eavesdropper's existence.

The proposed IoT quantum network contains $K$ quantum nodes (QN) and one central quantum server (QS) connected through an optical fiber. In order to share the entanglement and information between the QN and the QS. We assume that one quantum repeater (QR) exists between every $k$th QN and central QS, and the overall number of QRs equals $K$. In real-world problems, the number of QRs is larger than one. The distance between every $k$th QN and central QS is denoted by $d_k$, which is equal to or less than 20 km in our case. Note that the number of QRs between the QN and QS is strongly influenced by the distance and type of optical fiber used. The type of the adopted QR is the MIM quantum repeater protocol. The described IoT quantum network is shown in Figure 5.

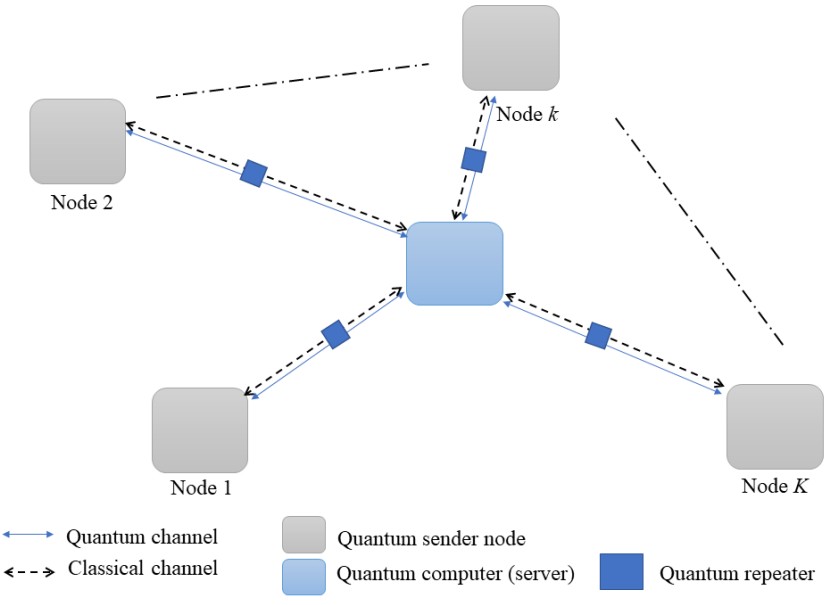

**Figure 5.** Structure of quantum network consists of *K*-nodes connecting to one quantum server through classical and quantum channels.

We consider that each QN requires local memory to send information and local control to perform entanglement with QS. We also supposed that a single qubit memory is sufficient for the QN (in most applications, one qubit memory is enough). The central QS can process and store the data. The function of each QN is specified according to the required application. The entanglement between QN and QS is shared based on the link layer protocol suggested by [29]. Each QN and QS consists of a single quantum network interface card (QNIC), which has a similar function to the network interface card (NIC). It is interesting to highlight that the QNIC allows QN to communicate with QS using the network stack. Each QR node contains two QNICs to share the entanglement between QN and QS and forward data from QN to the QS using optical fiber as a communication channel. We implemented the RuleSet suggested by [30], which coordinates the communication between QN and QS.

Now, we are in a position to define the CP for the IoT quantum network. The communication methodology between the $k$th QN and the central QS has two main steps: sharing entanglement and sending information (note that each step consists of sub-steps). The CP is provided in detail as follows:

First step: sharing entanglement

- The $k$th QN sends a qubit state to its corresponding QR in order to initiate entanglement generation.

- The BSA situated inside the $k$th QR performs some calculations according to the distance $d_k$, photon emission timing (time required by each node to emit photon), and photon recovery timing (an essential property of a single-photon detector represents the desired time to recover the formal efficiency of the detector).
- The $k$th QR forwards the result (the aforementioned calculations) to the $k$th QN and the central QS using the classical channel in order to establish the entanglement swapping via the quantum channel between the $k$th QN and the central QS.
- The central QS notifies the $k$th QN that the entanglement sharing was successful.

second step: sending information

- The $k$th QN forwards the information to the QS via the classical channel.
- When the QS receives the information, it sends an acknowledgment (ACK) signal to the $k$th QN. If the ACK does not reach the QN at a specific time, then the QN resends the information again.

Figure 6 illustrates the communication procedure between the QN (sender) and the central QS (receiver). It is important to highlight that the proposed IoT quantum network describes sending data from the QN to the central QS through the quantum and classical channels. These nodes share a few numbers of qubits that can be entangled and exchange quantum information using a quantum internet network. This protocol guarantees that eavesdroppers cannot reach the results of measured information due to applying entanglement before sending data [32].

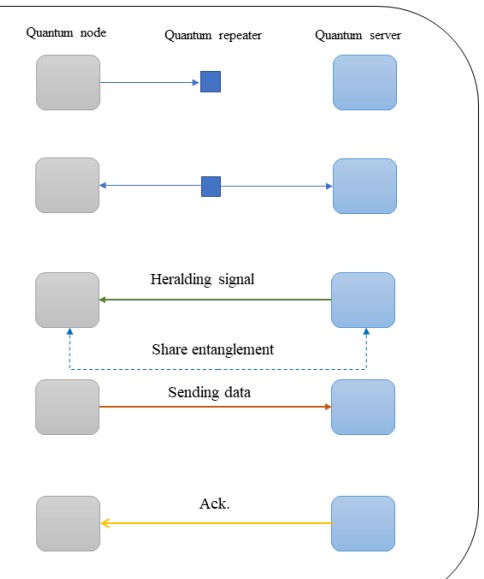

1. The sender node transmits a signal through the quantum channel to the quantum repeaters.

2. The quantum repeater shares the entanglement between the quantum node and the quantum server using entanglement swapping.

3. The quantum server sends a heralding signal to the quantum node which states that the entanglement sharing was successful in order to share data

4. Quantum node transmits quantum data through the quantum channel.

5. When the quantum server receives data, it forwards an acknowledgment to the quantum node. If the quantum node does not receive the acknowledgment at a specific time, then it will transmit the data again.

**Figure 6.** The proposed communication procedure between quantum node (sender) and quantum server (receiver).

## 5. Conclusions

This paper handles the problem of the security and processing speed of the IoT classical network and substitutes the traditional client–server model with a quantum one called the IoT quantum network. This new model connects the quantum nodes to the central quantum server via a quantum channel that contains a quantum MIM repeater. We also show the communication procedure steps for sharing data between a single quantum node and the central quantum server. In future work, we will focus on extending the proposed IoT quantum network to handle both the sending and receiving of data between the quantum client and quantum server and define their communication procedures. Finally, we will build extensive simulations for different IoT quantum network systems using QuISP.

**Author Contributions:** Conceptualization, D.S. and L.B.; methodology, D.S. and L.B.; writing—original draft preparation, D.S.; investigation, D.S.; writing—review and editing, L.B. supervision, L.B. All authors have read and agreed to the published version of the manuscript.

**Funding:** The research was supported by the Ministry of Culture and Innovation and the National Research, Development and Innovation Office within the Quantum Information National Laboratory of Hungary (Grant No. 2022-2.1.1-NL-2022-00004). L. Bacsardi thanks the support of the Janos Bolyai Research Scholarship of the Hungarian Academy of Sciences ( Grant No. BO/00118/20).

**Data Availability Statement:** Not applicable.

**Conflicts of Interest:** The authors declare no conflict of interest.

## Abbreviations

The following abbreviations are used in this manuscript:

| | |
|---|---|
| BQC | Blind Quantum Computing. |
| BSA | Bell State Analyzer. |
| CP | Communication Procedure. |
| CSS | Calderbank–Shor–Steanes. |
| IoT | Internet of Thing. |
| MIM | Meet In the Middle. |
| NIC | Classical Network Interface Card. |
| NLAs | Noiseless Linear Amplifier. |
| QKD | Quantum Key Distribution. |
| QN | Quantum Node. |
| QNIC | Quantum Network Interface Card. |
| QoS | Quality of Surface. |
| QR | Quantum Repeater. |
| QS | Quantum Server. |
| QuISP | Quantum internet Simulation Package. |

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
