# Peer review of "Using Quantum Nodes Connected via the Quantum Cloud to Perform IoT Quantum Network"

_condensedmatter, doi:10.3390/condmat8010024_

Round 1

Reviewer 1 Report

Using Quantum Nodes Connected via the Quantum Cloud to Perform IoT Quantum Network

Authors: Doaa Subhi, and Laszlo Bacsardi
In this work, Subhi et al.
introduced a novel Internet of Things (IoT) quantum network that provides high security and Quality of Service (QoS) compared with the traditional IoT network. The author claims that the new model connects the quantum nodes to the central quantum server via quantum channel that contains a quantum MIM repeater.
The manuscript is very well written, the investigation is thorough, and analytically and numerically are reported appropriately in detail. However, I feel there are some key points that the authors can add to the manuscript, from perspectives of design, supporting experiments, potential applications and system-level implementations, and context of prior art and the potential application to quantum
communication, before publication in Condensed Matter open access. These points are listed

The paper would greatly profit if the authors supplement all the results with clear references and in case a given reasoning is original indicate it as well.
1. The authors ignore some important reference in the introduction: on the use of entanglement to improve the overall stability and against fluctuations of the measurement settings in qubit sytems
J. Phys. B 42, 0335502 (2009),its experimental implementation https://doi.org/10.1103/PhysRevA.81.012305, in the presence of noise https://doi.org/10.1088/0031-8949/2010/T140/014062.

About recent works on how phase modulation of coherent states plays in quantum communication channels like in the paper https://journals.aps.org/pra/abstract/10.1103/PhysRevA.92.012317, https://doi.org/10.1088/0031-8949/2010/T140/014062 and the use of probabilistic noiseless linear amplifiers both at the encoding stage https://doi.org/10.1364/JOSAB.36.002938 where the information is coded on phase shifts and at the decoding stage https://journals.aps.org/pra/abstract/10.1103/PhysRevA.93.062315

Author Response

Dear sir/madam

I hope you are doing well.

Thank you for your comments.

Please find the response in the attachment.

Best regards

Doaa

Reviewer 2 Report

Please see attached PDF file.

Author Response

(The authors gave the same response as above.)

Round 2

Reviewer 1 Report

The new version of the manuscript addresses most of the worries and I am therefore recommending acceptance of the manuscript.

Reviewer 2 Report

The authors have addressed most of the comments. The English should still be carefully checked, as I spot several grammar issues of the paper. The paper might be accepted after the check.